# Clientelism, Turnout and Incumbents' Performance in Chilean Local Government Elections

**Mauricio Morales** [1,*] and **Fabián Belmar** [1,2,*]

1  School of Political Science and Public Administration, Faculty of Social and Juridical Sciences, Universidad de Talca, Santiago 8940583, Chile
2  School of Government, Universidad Adolfo Ibáñez, Santiago 7941169, Chile
*  Correspondence: mmoralesq@utalca.cl (M.M.); fabian.belmar@utalca.cl (F.B.)

**Abstract:** Parties and their leaders are linked programmatically and non-programmatically with citizens, incentivising the latter to vote in elections and seeking to influence their choices. In this paper, we analyse the effects of politician–voter linkages on the electoral performance of incumbent mayors in Chile and on electoral turnout in their municipalities. To measure the linkages, we use personal meetings that mayors hold with citizens. While some mayors use this mechanism to solve problems of general interest (programmatic meetings), others do so to provide bureaucratic advantages or benefits for their constituents (non-programmatic meetings). We use a database of 44,162 personal meetings aggregated from Chile's 345 municipalities. We argue that increases in the number of meetings positively impact electoral turnout and increase the chances of success for incumbent mayors when they compete for re-election. This effect is particularly significant in the case of electoral performance and the re-election of mayors in municipalities with high levels of rurality. Finally, we report that the meetings not only help mayors to link with their constituents but also help them to publicise their political work.

**Keywords:** local government; clientelism; lobby; electoral turnout; local elections





## 1. Introduction

Political parties choose their electoral strategies according to the territorial level at which each election takes place. While in presidential and legislative elections the parties strive to publicise their programmatic proposals, in subnational elections they are likely to develop non-programmatic links with their potential voters to a greater extent (Levitsky and Murillo 2005; Kitschelt and Wilkinson 2007; Kitschelt et al. 2010). In municipal elections, this non-programmatic linkage becomes stronger because the campaigns are more personalised and the candidates try to respond to the concrete demands of specific groups of voters (Schröter 2010; Szwarcberg 2015; Zapata Osorno 2016; Freidenberg 2017). In these local settings, the construction and development of mediation networks are common (Caciagli 1996; Barozet 2005; Arriagada 2013; Ortiz de Rozas 2017). This task is facilitated for incumbent candidates, who can use discretionary practices to benefit their voters through resource transfers, bureaucratic advantages, job offers, the concession of favours, etc. (Lodola 2005; Nazareno et al. 2006; Remmer 2007; Albertazzi 2014).

Studies on non-programmatic linkage—and clientelism as such—tend to relate this type of practice to an eventual payoff in electoral profits, especially for incumbents (Brusco et al. 2004; Desposato 2007; Singer 2009; Finan and Schechter 2012; Calvo and Murillo 2013; Gonzalez Ocantos et al. 2014). In the local arena, for example, incumbents control municipalities' economic resources and can distribute benefits, thus gaining an advantage over eventual challenger candidates. We know that incumbents enjoy the benefits of the post and are better placed than a challenger to build networks of intermediaries. However, we have less evidence about how this type of clientelist practice impacts citizens' electoral

turnout (Nichter 2008; Gans-Morse et al. 2009; Szwarcberg 2012) and incumbents' electoral performance (Zittel 2015; Núñez 2018; Lucas 2021).

Focusing on the case of Chile, in this paper, we propose a novel method to measure the patron–client link. We then estimate the impact of this mediation on the electoral performance of incumbents and on electoral turnout. To measure the patron–client link, we used the personal meetings held between mayors and citizens—technically known as *audiencias* in Spanish—in Chile, a country that since 2015 has regulated lobbying and standardised the influence that citizens have on the authorities' decisions on issues of public interest. In principle, these personal meetings must be granted for topics that affect the general interest, so they are not intended to resolve issues that involve citizens' personal interests (non-programmatic meetings). Such matters of general interest refer to the implementation of policies, plans or programmes (programmatic meetings). However, according to the supervisory body Council for Transparency (CPLT, by its acronym in Spanish), these meetings have been used for the personal management of problems, with the mayors resorting to a discretionary use of the tool that involves conceding bureaucratic advantages to certain citizens.

To study the relationship between personal meetings and the electoral performance of incumbents and electoral turnout, we distinguish between programmatic and non-programmatic meetings. We categorise as programmatic meetings those that respond to matters of general interest. Non-programmatic personal meetings are those aimed at solving particular problems of the applicant. The personal meetings, of course, show only a fragment of the personalised relationship between the mayor and citizens. While we recognise this limitation, we suggest that by using formal mechanisms, these data bring us closer to a rather more specific measurement of the link between representatives and constituents. The other advantage is that the data are available for all 345 municipalities of the country. We counted 44,162 personal meetings, all downloaded from the portal www.infolobby.cl (accessed on 21 March 2021). The question guiding this research was: to what extent do variations in the volume of personal meetings explain electoral turnout and the electoral performance of incumbent mayors?

The text is divided into five sections. First, we review the main postulates regarding the link between politicians and voters. Second, we describe the linkage types between them, differentiating between programmatic and non-programmatic linkages. In addition, we identify the proxies that have been used to measure clientelism, and based on this review, we propose our own. Third, we present the hypotheses, method and results of the data analysis. Fourth, we close the document with our main conclusions.

## 2. Party–Elector Linkage and Electoral Behaviour

There is an extensive literature on how parties and their leaders link with voters (Kitschelt et al. 2010; Luna 2010, 2014; Calvo and Murillo 2013; Morgan and Meléndez 2016; Vommaro and Combes 2016; Freidenberg 2017). Kitschelt (2000) identified three ideal types of relationship between representatives and constituents: programmatic, charismatic and clientelist. In the case of programmatic linkages, voters choose their candidates based on public policy proposals. In charismatic linkages, voters select their preferences according to their characteristics. In a relationship deemed to be clientelist, voters choose their preferences based on a material exchange with the candidate or representative. This exchange can be based on goods, services, bureaucratic advantages, money or public jobs, among other benefits.

Subsequently, this typology was reformulated with the development of mediation strategies and more-complex electoral campaigns. The most important finding was that in some cases the different types of linkage could coexist in the same party strategy (Kitschelt and Wilkinson 2007; Kitschelt et al. 2010; Luna 2014). In other words, parties and their leaders can develop different types of links depending on the voter they seek to convince. Thus, linkage types are not mutually exclusive. Parties could, for example, link programmatically

with their staunchest voters and develop clientelist links with those who are disaffected or do not even share their ideological position.

On the latter point, academics have focused on how these clientelist links are built, and especially on how networks of intermediaries are set up (Auyero and Güneş-Ayata 1997; Durston 1999; Auyero 2001; Durston 2005a; Combes 2011; Vommaro and Combes 2016). However, less space has been devoted to analysing how these networks affect the electoral success of those who lead them. In the case of incumbents, in particular, they have resources and also discretionary power to distribute benefits among those they govern. Such networks of intermediaries, for example, can serve to grant bureaucratic advantages or simply to hand out goods to those who request them. In the case of Chile, Valenzuela (1977) and Durston (2005b) studied the role of so-called mediators or brokers, while Barozet (2005, 2012); Valenzuela-Van Treek and Arévalo (2015); Vallejos (2016); Pérez Contreras and Verón (2018); and Contreras (2020) analysed the mechanisms of control and co-optation used by mayors to gain a reputation as local *caudillos* or *caciques* and obtain electoral returns.

The literature on clientelism that has addressed the effects of these practices usually revolves around vote buying (Desposato 2002; Brusco et al. 2004; Schaffer and Schedler 2007; Schröter 2010; Stokes et al. 2013; Lisoni 2018). As we will see later, this practice has attracted academic interest because it allows those in power to buy voters' choices and thus guarantee favourable votes. As Stokes et al. (2013) indicate, these mechanisms have lost effectiveness as reforms for the transparency of electoral processes and polling booth secrecy have gained traction, generating an increasingly less opaque vote. However, due to the refinement of research methods, an incipient literature has emerged that maintains that the harvesting of votes through clientelist practices is limited (Kramon 2016; Núñez 2018; Cantú 2019). These studies pay attention to the electoral advantage obtained by purchasing votes. They conclude that transfers or direct benefits are not very effective compared to other types of private distributions such as constituency service or patronage.

Although most of the literature on the effects of personalised linkage focuses on electoral advantages, some researchers have investigated its consequences for participation (turnout buying). This is due to the informality of these practices and the inability to monitor voters, preventing researchers from measuring their effects on voting and electoral success. For example, the work of Nichter (2008) elaborates on the payback from these practices, arguing that in addition to their effects on electoral preferences, consequences can also be observed on turnout. The benefits granted, in other words, would not encourage voting choice, but rather attendance at the polls. Furthermore, the author points out that several vote buying studies (e.g., Stokes 2005) actually involved turnout buying, as in some cases the orientation of distributed benefits does not consider mechanisms for monitoring the vote.

In a later study, Gans-Morse et al. (2009) argue that when it is not possible to oversee voter preferences, politicians can use other strategies or even combine them, notably vote buying, turnout buying, persuading disaffected voters or opponents (double persuasion), encouraging opponents to abstain (negative turnout buying) and rewarding supporters (rewarding loyalists). Stokes et al. (2013) point out that the turnout buying model is particularly important where voting is voluntary, as it provides a basis for capturing the votes of ideologically sympathetic voters. They add that the effect of this practice is crucial in scenarios of low turnout because—although it is important to persuade independents— electoral success does not depend on convincing new voters but on getting loyal voters to the polls.

## 3. Approaches to the Operationalisation of Clientelism

Clientelism is a polysemic concept and an analytic category subject to constant disciplinary discussion. From the perspective of political science, it has been studied through the analysis of the preferential or discretionary use of resources and the co-optation of jobs in government departments (Rehren 2000; Gordin 2006; Giraudy 2007; Moriconi Bezerra 2011; Oliveros 2016). Prominent works have focused on the particular handling of transfers

from central power to local governments (Valenzuela 1977; Acuña et al. 2017; Corvalan et al. 2018). However, also important are those that examine municipal spending, whether targeted transfers, a discretionary increase in personnel or the management of social plans (Gordin 2006; Moriconi Bezerra 2011; Mimica and Navia 2019). In general terms, these studies have historically associated clientelism with corruption (Scott 1969; Kurer 1993; Rehren 2000; Lehoucq 2003) and as such considered it a symptom of the deterioration of a democratic regime.

To develop our proposal for measuring non-programmatic clientelist linkages, we will first review how previous studies have operationalised the concept based on quantitative empirical analysis. First, following the work of Schröter (2010), we identify the main clientelist phenomena that have been studied in the academy. Following that, guided by the work of Hicken (2011), we associate the main operationalisations carried out as part of the empirical analysis of clientelism with a specific phenomenon of study. Table 1 summarises this exercise.

**Table 1.** Operationalisation of clientelism in the social sciences.

| Phenomenon | Dimension | Operationalisation/Measurement | Noted Studies |
|---|---|---|---|
| **Vote buying** | Exchange characteristics of the patron–client relationship | Opinion surveys and interviews with clients and patrons | Brusco et al. 2004; Finan and Schechter 2012; Nichter 2014 |
| | Party efforts to deliver benefits | Perception surveys of clients | Singer 2009; Kitschelt et al. 2010; Gonzalez Ocantos et al. 2014 |
| **Constituency service** | Candidates' economic capacity | Electoral budget and spending | Keefer and Vlaicu 2008; Kitschelt et al. 2010 |
| | Advantage of incumbency | Incumbency and periods in the post | King 1991; Gardner 1991 |
| | Personalisation of politics | Absence of programmatic parties | Cruz and Keefer 2010; Zittel 2015 |
| **Mediation networks** | Sociodemographic characteristics of voters | Levels of rurality and poverty | Desposato 2007; Landini 2013 |
| | Economic capacity of parties | Spending on electoral campaigns | Kitschelt et al. 2010 |
| | Construction of networks of intermediaries | Identification of intermediaries or brokers | Crossley 2010; Larreguy et al. 2016 |
| **Patronage** | Level of corruption of institutions | Perception of corruption | Persson et al. 2003; Keefer 2007 |
| | Size of public investment/construction budgets | Size of investment in projects | Keefer 2007 |
| | Capture and instrumentalisation of institutions | Size of the public sector (wages and/or personnel) | Remmer 2007; Robinson and Verdier 2013 |

Source: prepared by the authors based on Schröter (2010) and Hicken (2011).

The most classical empirical studies on clientelism have investigated vote buying. Research into the exchange of favours, goods or services for electoral support and votes has developed around the operationalisation of two areas. Some scholars have investigated the characteristics of the exchange, identifying the favours or goods offered by a patron and the latter's relationship with the client (Brusco et al. 2004; Finan and Schechter 2012; Nichter 2014). Using surveys, others have measured parties' efforts to distribute benefits to clients (Singer 2009; Kitschelt et al. 2010; Gonzalez Ocantos et al. 2014).

A second group of authors have addressed clientelism from the point of view of constituency service, that is, how the patron provides services to the community, specifically

to his constituents. The services negotiated can be the provision of basic services such as public lighting, street and pavement repairs, etc. Operationalising this practice has contributed to a better understanding of the congruence between parties and voters in their programmatic preferences (Cain et al. 1984; Kitschelt et al. 2010; André et al. 2014); politicians' economic capacity (Keefer and Vlaicu 2008; Kitschelt et al. 2010); the advantage of incumbency (Gardner 1991; King 1991); and the degree to which politics has become personalised (Cruz and Keefer 2010; Zittel 2015).

A third group has studied how networks of intermediaries involved in the granting of favours are constructed. It has been suggested that sociodemographic variables are important when identifying the presence of patronage networks, which increase in the lowest-income and most vulnerable sectors (Desposato 2007; Landini 2013). Given that the ability to deploy networks of intermediaries depends on the parties' economic capacity, much of the literature focuses on studying their budgets and spending (Kitschelt et al. 2010).

A final group of studies has analysed patronage. In this clientelist dynamic, what the patron offers consists of public resources, such as employment in government departments or selective spending on projects, policies or programmes that benefit a specific group. This phenomenon is usually studied using theoretical frameworks of corruption (Kurer 1993; Roniger 2004; Keefer 2007) and of government variables such as the size of the public sector and spending on personnel (Remmer 2007; Robinson and Verdier 2013; Mimica and Navia 2019).

The operationalisation explored in this work focuses on a discretionary use of those mechanisms or tools that the institutions provide to link with voters. However, we emphasise that—unlike the bulk of the literature, whose concern is with informal institutions—our interest is in formal mediating mechanisms. In this approach, it is common for researchers to focus their studies on the efforts of political machines to generate and strengthen this link. Indeed, (e.g., Auyero 2001, 2002, 2011) mentions the importance and difficulty of establishing a link based on the "foundational favour"—i.e., the first meeting between the patron or intermediary and the client, who, in turn, initiates the relationship. In our case, and because personal meetings are requested by the citizens themselves and take place in the same municipality, these associated costs do not exist.

Specifically, we maintain that the meetings (*audiencias*) granted by mayors constitute patronage practices because they involve the "capture" of institutions in order to favour some citizens with personalised attention or solutions to their problems. We argue that the institution of the personal meeting has been "captured" because it is used discretionarily by these politicians. As a formal mediation mechanism established by law, it may only be used for matters of public interest (i.e., the mayor should not attend to requests of a personal nature). Every time mayors accept these personalised meetings, they exceed their mandates, as they ignore that the municipalities have established mechanisms to meet these citizens' requirements. Of course, not all meetings can be considered clientelist because some are adjusted to the law and address issues of interest to the community.

To the exploitation of the institution is also added the distribution of benefits because if some citizens are invited to the meeting but others are not, it means that the mayors are arbitrarily distributing targeted clientelist benefits (Belmar and Morales 2020). Even if we do not know if the mayor actually delivered the requested benefits, what we can be certain of is that the citizens received preferential attention. While some citizens have to adhere to the established procedures to be heard in the "attention to the public" office of some departments of the municipality, others avoid these bureaucratic procedures by requesting a personal meeting and being heard directly by the mayor himself. This is one more reason to maintain that, from the moment the mayor grants an off-grid meeting of this sort, he or she is engaging in a clientelist practice.

The discretionary and clientelist (none of the above) way in which the *audiencia* mechanism has been used has even been covered by the press at both the local and national levels. While it is not within the objectives of this paper to carry out an exhaustive review of the press archives dealing with reactions to the capture of this institution, we believe it is useful to mention some illustrative cases. Below are some examples (links to the press

evidence we present can be found in the bibliography). Early on, the supervisory body itself noticed the volume of meetings granted by mayors shortly after the implementation of the law regulating lobbying and meetings in municipalities (CPLT 2015). On this occasion, the CPLT reported that mayors are the politicians who have met the most with citizens to mediate interests, a behaviour that has been maintained for almost a decade following the enactment of this law. The media quickly echoed the details of these meetings (non-programmatic distribution demands) and even covered the donations that citizens make to the authorities when attending meetings in mayors' offices (Zamorano 2016).

A piece of background information that we think reinforces our argument is how mayors refer to the use they make of the meetings. Moreover, some have mentioned the number of personal meetings they grant as a sign of their strong commitment to their constituents and the transparency of their activities. Consequently, the media has reported on these declarations, giving an overview of these mayors' performance to the entire population. In other words, mayors are aware that holding a large number of meetings is an advantage they can capitalise on and use to obtain visibility in the press. To illustrate this, we can use the case of the authority that most used the *audiencia* mechanism during our period under analysis, the mayor of the municipality of Teno. This small commune in the central part of the country received the attention of local and national media for the management of its mayor, highlighting her for holding more meetings than even the National Congress of Chile (Wilson 2016), which at the time was composed of 120 deputies and 38 senators. In other words, the mayor of a commune with fewer than 30,000 inhabitants scheduled—in the same period—more meetings than the representatives of more than 17 million Chileans.

Another similar example to that of the mayor of Teno is that of the mayor of San Esteban, an even smaller commune with fewer than 20,000 inhabitants. In this case, local media again highlighted the large volume of meetings that the mayor held, receiving citizens in the municipality's offices (Ríos 2016). In both cases, the municipalities' web portals and social networks were responsible for popularising this news in the media. National media also reported that, for the 2016 elections, the mayors who gave the most meetings were re-elected and even had full reports dedicated to their profiles and their reasons for giving so many meetings (Bravo and Wilson 2016). Eventually, the media made visible the complaints of academics and corruption specialists about the inadequate use of the law and the lack of oversight and fines associated with its violation (Gómez 2020).

## 4. Hypotheses

Unlike the currents listed in Table 1, our proposal focuses on measuring the personalised relationships between mayors and their constituents established through formal channels, those being personal meetings (specifically, personal meetings of an irregular nature). Based on the preceding theoretical discussion, we propose the following hypotheses.

**H1.** *Mayors who granted a greater number of personal meetings obtained a greater percentage of votes.*

As we saw in our review, an extensive literature maintains that the aim of personalised linkage strategies is to obtain electoral gains. However, the absence of specific data on this type of relationship makes it difficult to use statistical methods. In our case, we have resorted to official sources of information that provide data on more than 44,000 personal meetings for the period under study. In this way, we will be able to estimate the impact that the number of personal meetings held has on the share of votes obtained by incumbents.

**H2.** *Mayors who granted a greater number of personal meetings achieved re-election.*

Incumbents enjoy several benefits associated with holding office, including granting personal meetings to their constituents in order to resolve their particular problems. However, the evidence shows that, although incumbents do have advantages when facing an election, these do not ensure that they will necessarily hold on to their post. In other words,

an increase in the number of personal meetings granted by mayors could increase their share of the vote without guaranteeing their re-election. Indeed, even when an incumbent can set up networks of intermediaries in a municipality, poor management of those links (i.e., if they do not satisfy voters' demands) could harm them.

**H3.** *Municipalities that record a greater number of personal meetings have higher electoral turnout.*

Studies on non-programmatic linkages and clientelist practices are usually related to political co-optation and vote buying. However, due to the impossibility of verifying that support for citizens is actually rewarded, part of the literature focuses on studying the effects on electoral turnout. Personalised and discretionary linkages undoubtedly help bridge the gap between the authorities and the citizenry. In this scenario, it is not surprising that a mayor who presents a governance strategy that is much more open to the community (due to measures such as interacting with citizens through personal meetings) generates greater incentives for residents of the municipality to participate actively in political life.

## 5. Method

To measure non-programmatic linkages, we use the number of programmatic and non-programmatic personal meetings granted by each mayor of Chile's 345 municipalities. It is assumed that the mechanism of the personal meeting is intended only to resolve matters of general interest. That is what the law understands as "regular meetings" (or programmatic meetings, according to our conceptualisation). Personal meetings granted to resolve matters of interest to individuals or groups (as opposed to the community as a whole), which are not provided for in the law, we classify as "irregular" personal meetings (or non-programmatic meetings, according to our conceptualisation). What is surprising is that nearly 85% of all audiences are classified in this category. Given that municipalities in Chile have a high population variance, we also constructed a "personal meetings rate" for every thousand inhabitants of each municipality, considering the total number of personal meetings granted for the period of analysis.

Of course, our conceptualisation could have problems in correctly identifying the various topics for which these personal meetings are requested. However, that is not the intention of our work. In fact, this would be a problem if this mechanism had been established to resolve constituency service matters (i.e., personal and community requirements of constituents), as that would make it necessary to categorise the personal meetings according to their content. On the contrary, and because the law is quite clear on this point, we consider that irregular personal meetings involve clientelist practices because by allowing them to happen, the mayor is discretionally granting bureaucratic advantages.

We examined the personal meetings granted by mayors during the pre-electoral period of the 2016 municipal elections, between the months of October 2015 and October 2016. We worked with the official records of personal meetings granted by the mayors, corresponding to the public agendas of the incumbent mayors of the country's 345 municipalities, available on the portal www.infolobby.cl (accessed on 21 March 2021). In total, 44,162 personal meetings were analysed.

We seek to determine the effect of the mayors' "personal meetings rate" on three variables. The first variable is the electoral performance of mayors seeking re-election, measured as their percentage of the vote. The second variable is the success or failure of the mayors, measured with a dummy variable that indicates whether the mayor was re-elected or not. The third variable is electoral turnout, measured by the votes cast as a percentage of the electoral register for 2016.

Turning to the independent variables, they include campaign spending (more specifically, the difference between the spending of the incumbent in 2012 and in 2016); the number of periods that each mayor has held office; the party coalition to which each mayor belongs; and the sociodemographic characteristics of the municipalities, particularly rurality and poverty. We use linear regression models when the dependent variable is

quantitative and probit regression models when the dependent variable is dichotomous. Given the wide population variance of the country, the models were weighted according to the votes validly cast in the 2016 mayoral election—in the cases of the percentage of the vote and re-election of mayors—and by the population that is of voting age for the analysis of electoral turnout.

### 6. Data and Discussion of the Results

The mechanism through which mayors meet with citizens is known as a personal meeting, established after the enactment of Law No. 20730, which regulates lobbying and the representation of private interests. In the terms of this law, a personal meeting is defined as one in which an "active subject" receives a "passive subject" to hear his or her requests or requirements. A public authority, in this case the mayor, is considered to be the active subject, while the lobbyist or interest broker is considered to be the passive subject. It is of the utmost importance to emphasise that the personal meetings covered by this law may only be scheduled to deal with matters regulated by the law.

In order for the personal meetings to be conducted correctly, they must necessarily be limited to one or another of these matters. Control over compliance with the regulations is the responsibility of the CPLT. However, the CLPT cannot decide whether the personal meetings were granted correctly. This is the job of the municipality's transparency officer, who is in charge of scheduling, registering and classifying the personal meetings according to their subject matter. In principle, requests that do not comply with the law should not be attended to by the mayor. In the event that the active subjects ignore the regulations and grant personal meetings that have been requested on some issue that does not correspond to any of the matters indicated in the law, the personal meetings will be declared as falling under "none of the above".

Despite the fact that the matters for which citizens can request personal meetings are related to or pursue purposes of public interest, the data show that the authorities at the local level (mayors) have not followed this principle in implementing the law. Figure 1 shows the number of personal meetings scheduled in the pre-election period from October 2015 to October 2016. Although it is to be expected that there will be requests that cannot be classified as falling under any of the matters indicated in the law, these should be the minority. However, more than 85% of the personal meetings granted in the 2016 pre-electoral period were personal meetings that did not conform to any of the matters envisaged by the law.

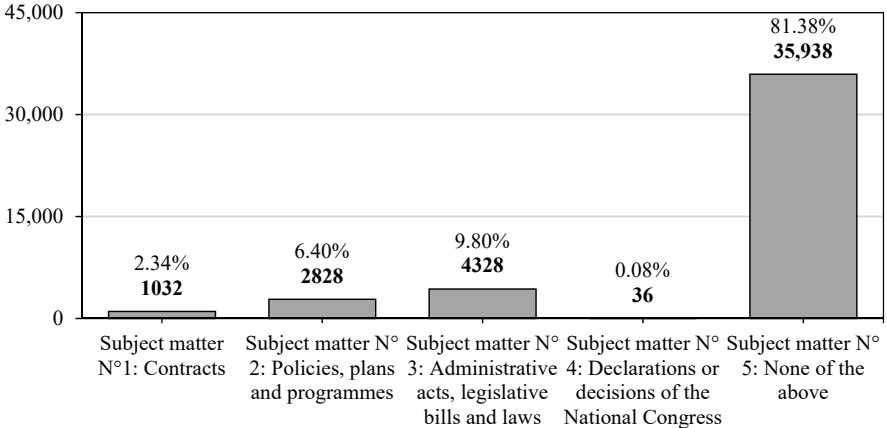

**Figure 1.** Number of personal meetings granted by mayors by subject matter, 2015–2016. Source: prepared by the authors with data from www.infolobby.cl (accessed on 21 March 2021).

The number of these personal meetings and the subject matters for which they are granted, moreover, are entirely at the mayors' discretion. In line with the results of Belmar and Morales (2020), it can be seen that both the number of personal meetings and their

content depend on the electoral cycle. While in inter-electoral periods the personal meetings granted stick to the matters indicated by the law, in pre-electoral years, and as the election approaches, the subject matter of the personal meetings becomes personalised. That is, the requests change from being programmatic (plans, policies and programs) to being actions affecting particular interests (individualised social benefits) that do not affect the public interest

To test our hypotheses, we decided to build a series of statistical models. The independent variables that we include in the models are the following:

1. *Percentage of vote in the previous election.* Naturally, one of the main predictors of the percentage of the vote that an incumbent will obtain is how the incumbent performed in the previous election;
2. *Personal meetings rate.* Studies indicate that linkage with voters is a determinant of electoral success. We include the number of personal meetings granted by mayors per thousand people (the personal meetings rate), also adding its quadratic and cubic in order to identify possible non-linear patterns;
3. *Experience in office.* The most experienced mayors may not need formal mechanisms because they know the territory. We measure experience by the number of terms the mayor has held office;
4. *Political affiliation.* To evaluate possible partisan-programmatic effects, we differentiate between the parties of the New Majority (Centre-Left), to which we assign a value of 1, and the rest of the parties, to which we give a value of 0;
5. *Incumbent expenditure.* Guided by previous studies in Chile (e.g., Morales Quiroga and Rodríguez 2010), we include mayors' campaign expenditure. To measure spending, we used the formula proposed by Acevedo and Navia (2015): the percentage that each candidate's spending represents based on the spending limit established by law for each municipality. Thus, for example, if the spending limit is 100 and a mayoral candidate spent 50, his spending corresponds to 50% of the limit;
6. *Increase in expenditure,* 2012–2016. We add a temporary spending control, calculating the difference in spending of an incumbent from one election to another. Incumbents can increase or reduce their spending from one election to the next, which can have an impact on their electoral performance;
7. *Difference in expenditure.* To capture the competition in the election, we used the spending of incumbents and challengers in the 2016 election. For this we calculated the difference between the incumbent and his main contender—that is, the person who came closest in vote percentage.

We include as control variables the sociodemographic characteristics of the municipalities in terms of poverty and rurality.

Table 2 shows the models. Influenced by the literature on clientelism reviewed in the theoretical section, we favoured a strategy of segmenting municipalities by combinations of rurality and poverty. First, we calculated the averages and the standard deviations of rurality and poverty. Next, we divided municipalities into four groups, starting with those that are both rural and poor. We included in a first group those municipalities that are one standard deviation above the average of rurality (over 32%) and of poverty (over 23.1%). A second group included those with rurality over 32% but poverty below 23.1%. In a third group we placed municipalities with a lesser rural population (less than 32%) but with poverty over 23.1%. Finally, in the fourth group we placed municipalities that are low in both rurality and poverty (below 32% and 23.1%, respectively).

**Table 2.** Predictors of mayors' percentage of the vote (2016).

|  | Total | (1) +Rural & +Poor | (2) +Rural & −Poor | (3) −Rural & +Poor | (4) −Rural & −Poor |
|---|---|---|---|---|---|
| Percentage of the vote 2012 | 0.625 *** | 0.115 * | 0.772 *** | 0.715 ** | 0.649 *** |
|  | (0.0918) | (0.159) | (0.175) | (0.258) | (0.160) |
| Personal meetings | 0.106 | −0.0743 | 1.097 ** | 0.207 | 1.033 |
|  | (0.255) | (0.198) | (0.597) | (4.848) | (1.319) |
| Personal meetings | −0.00101 | 0.00121 | −0.0313 * | −0.0496 | −0.0669 |
|  | (0.00369) | (0.00260) | (0.0184) | (0.787) | (0.0954) |
| Personal meetings | $2.50 \times 10^{-6}$ | $-2.61 \times 10^{-6}$ | 0.000199 * | 0.000860 | 0.000817 |
|  | $(9.63 \times 10^{-6})$ | $(6.45 \times 10^{-6})$ | (0.000132) | (0.0318) | (0.00125) |
| No. of terms in office | −2.149 *** | −1.678 | −1.495 | −2.818 | −2.106 * |
|  | (0.660) | (1.093) | (1.499) | (4.784) | (1.098) |
| Party (1 = N. Majority/0 = Other) | −6.639 *** | −2.353 | −2.792 | 4.390 | −7.951 *** |
|  | (1.680) | (2.553) | (3.513) | (5.608) | (2.883) |
| Incumbent expenditure | −0.0630 | 0.0686 | −0.241 *** | −0.0893 | −0.0608 |
|  | (0.0529) | (0.0850) | (0.0888) | (0.207) | (0.0927) |
| Increase in expenditure | 0.0678 ** | −0.0591 | 0.0735 | 0.122 | 0.0595 |
|  | (0.0382) | (0.0519) | (0.0812) | (0.226) | (0.0667) |
| Difference in expenditure | 0.181 *** | 0.234 *** | 0.104 | 0.299 * | 0.195 ** |
|  | (0.0421) | (0.0694) | (0.0691) | (0.164) | (0.0752) |
| Constant | 25.50 *** | 43.66 *** | 28.59 ** | 16.51 | 23.60 *** |
|  | (5.101) | (9.331) | (13.24) | (12.37) | (8.778) |
| Observations | 263 | 86 | 51 | 23 | 103 |
| R-squared | 0.267 | 0.314 | 0.541 | 0.722 | 0.259 |

Standard errors in parentheses. *** $p < 0.01$, ** $p < 0.05$, * $p < 0.1$. Source: prepared by the authors with data from www.servel.cl and www.infolobby.cl (accessed on 21 March 2021).

The results of the models are in line with theoretical expectations and partially support our Hypothesis 1. In the first place, the general model shows that the main predictors of incumbents' electoral success are campaign spending—those mayors who increased campaign spending compared to the previous election were more successful—and the vote achieved in the previous election: the better the mayors performed in the 2012 elections, the better their performance in those of 2016. Furthermore, the difference between the incumbent's and challenger's spending is also significant, with better results being obtained as the spending gap increases. In this model, personal meetings do not have a statistically significant effect. These results are the same for all the groups except for municipalities with above average rurality and below average poverty (the second group described above). Here we see a very particular effect of the personal meetings on the mayors' percentage of the vote. While increases in the personal meetings rate favour the percentage of the vote obtained by these mayors, there comes a point where that support begins to decline (seen in the negative sign of the quadratic) and finally rebounds, as seen in the positive sign of the cubic of the personal meetings rate.

What does this result tell us? Both poverty and rurality are variables that have been widely studied in the literature on non-programmatic linkages. Rurality is a decisive factor because in these social contexts, a closer link is forged between authorities and citizens (Valenzuela 1977; Wolf 1980; Durston 1999; Corzo 2002; Lawson and Greene 2014; Vommaro and Combes 2016). In the Chilean case, rurality can be expected to have greater weight in the model because the link established in rural environments is usually more personalised and may include affective ties of reciprocity and familiarity (Durston and Miranda 2001; Durston 2005b; Moya and Paillama 2017; Luján Verón 2019). The theory indicates that voters and those they elect will likely have had links before participating in politics and will continue to have them even if the representative loses his position (Auyero 2001; Salazar 2019). That is, the relationship originates from a prior life in the community, transcending electoral periods.

The effect is pronounced in the case of rural municipalities with low levels of poverty because with a smaller gap in wealth, the asymmetry of power between the authority and the citizen is less acute. A case-by-case analysis of the records shows that in these municipalities personal meetings are granted in large part for matters that include the transfer of public resources for projects and applications for jobs in the municipality. This type of request is typical of clientelist practices. In addition, in this group, a capture of

government could eventually be involved, as citizens' requests are related to benefits and state resources that, if those requests were attended to, would be selectively distributed among those who are closest to the figure of the mayor.

Given that the results of the models show that the personal meetings rate has diminishing marginal returns, we constructed a graph with the predicted values to visualise the behaviour of the variable. Figure 2 shows the predicted values of the dependent variable according to the personal meetings rate. As we have been seeking to show, at first the relationship between the personal meetings rate and the percentage of votes is positive, with an increasing electoral payoff as more personal meetings are granted. However, the relationship between the variables then becomes negative, with the personal meetings ceasing to benefit the mayors electorally. Finally, and with the granting of a large number of personal meetings, the weight of the personal meetings once again becomes decisive, even though their effect on votes is diminished.

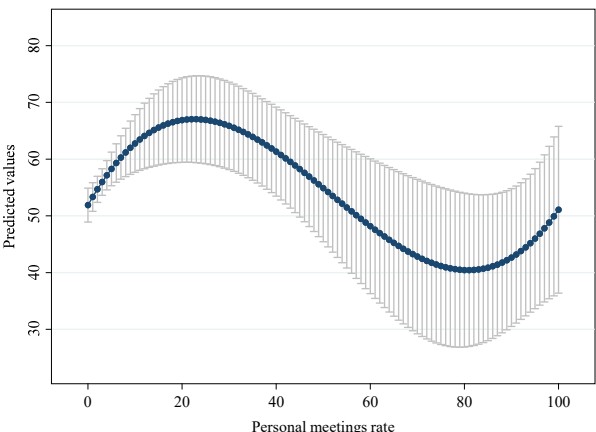

**Figure 2.** Predicted values for percentage of the vote according to personal meetings rate. Source: prepared by the authors with data from www.servel.cl and www.infolobby.cl (accessed on 21 March 2021).

As mentioned, we use these polynomials because both the press and previous studies have addressed how large numbers of meetings benefit mayors through visibility and may even have an effect on their re-election. Our explanation of this phenomenon is as follows. First, a positive relationship between the personal meetings rate and the votes obtained is generated because citizens would begin to perceive the mayor's administration positively—as mentioned in the press—compared to those who do not grant personal meetings, due to its developing an open-door policy. It is very likely that this mechanism helps create a milestone at the beginning of the relationship. Second, this relationship decreases (quadratic) because those mayors who grant more personal meetings are very likely to be unable to maintain personalised attention and thus delegate these meetings to other municipal officials or do not comply with some requests. This could certainly annoy citizens. In fact, several records of more populated municipalities make note of occasions in which, although an audience was requested with the mayor, the person who attended was the municipal secretary. Third, the (cubic) rebound in the relationship that we see when a greater number of personal meetings is granted is surely because once this threshold in the personal meetings rate has been crossed, there is growing media interest in the mayor's actions. As we have seen, both local and regional media start to highlight his work and his interest in meeting with the community, which reinforces the positive effect on voters' perceptions.

Our Hypothesis 2 suggests that the rate of personal meetings granted by mayors is decisive for their re-election. Following the same strategy as above, to test this we built a set of probit models. We assume that even though personal meetings are not significant as an explanation of the variation in incumbents' percentage of the vote in municipalities taken as a whole, there might be some effect on their chances of re-election. We note that

the percentage of votes is not necessarily an indicator of electoral success as measured by re-election. An incumbent can increase his percentage of votes and be defeated. Alternatively, he can lose votes but keep the seat. The results of this exercise are presented in Table 3.

**Table 3.** Predictors of mayors' re-election (2016).

| | **Total** | **(1)** +**Rural** & +**Poor** | **(2)** +**Rural** & −**Poor** | **(3)** −**Rural** & +**Poor** | **(4)** −**Rural** & −**Poor** |
|---|---|---|---|---|---|
| Percentage of the vote 2012 | 0.0486 *** | 0.00630 | 0.0812 ** | −0.180 | 0.0726 *** |
| | (0.0145) | (0.0220) | (0.0404) | (0) | (0.0216) |
| Personal meetings | 0.0469 | −0.0399 | 0.405 ** | 34.16 | 0.915 |
| | (0.0510) | (0.0311) | (0.174) | (0) | (0.622) |
| Personal meetings | −0.00136 | 0.000576 | −0.0141 ** | −4.474 | −0.235 |
| | (0.00144) | (0.000453) | (0.00596) | (0) | (0.174) |
| Personal meetings | $9.32 \times 10^{-6}$ | $-1.35 \times 10^{-6}$ | 0.000116 ** | 0.135 | 0.0164 |
| | $(9.56 \times 10^{-6})$ | $(1.14 \times 10^{-6})$ | $(4.85 \times 10^{-5})$ | (0) | (0.0116) |
| No. of terms in office | −0.201 ** | 0.189 | −0.752 *** | 9.418 | −0.259 ** |
| | (0.0997) | (0.152) | (0.228) | (0) | (0.121) |
| Party (1 = N. Majority/0 = Other) | −0.506 | −0.405 | −1.190 ** | 3.438 | −0.626 |
| | (0.342) | (0.417) | (0.514) | (0) | (0.448) |
| Incumbent expenditure | −0.0118 | 0.0118 | −0.0159 | −2.743 | −0.0142 |
| | (0.00865) | (0.0117) | (0.0151) | (0) | (0.0124) |
| Increase in expenditure | 0.0109 | −0.00929 | 0.000562 | 1.002 | 0.0135 |
| | (0.00670) | (0.00804) | (0.0160) | (0) | (0.00885) |
| Difference in expenditure | 0.0147 ** | 0.0225 ** | 0.0288 *** | 3.618 | 0.0127 |
| | (0.00625) | (0.0103) | (0.0102) | (0) | (0.00914) |
| Constant | −0.799 | −0.458 | −1.411 | 113.2 | −1.949 |
| | (0.836) | (1.449) | (2.456) | (0) | (1.212) |
| Observations | 262 | 85 | 51 | 23 | 103 |

Standard errors in parentheses. *** $p < 0.01$, ** $p < 0.05$. Source: Prepared by the authors with data from www.servel.cl and www.infolobby.cl (accessed on 21 March 2021).

The models show that personal meetings do not affect the mayors' re-election success significantly. However, in the case of Group 2, the model assumes that personal meetings would have a cubic effect on the probability of success. In other words, the greater the number of personal meetings, the greater the probability of a mayor being re-elected, an impact that diminishes once a certain level is reached but increases again if the mayor grants more personal meetings. Although no significance of the variables is observed for Group 1, the application of an F-test to capture significance shows a value of 0.001, so the variables follow the same behaviour as in the case of Group 2. To show these results more clearly, we plot Figure 3 in which we estimate the predicted values of the dependent variable (the probability that the mayor is re-elected) as a function of the independent variables and, especially, of the personal meetings. We take the rest of the variables at their averages and segment the results by groups of municipalities based on poverty and rurality, as in the previous model. We exclude Group 3 (−rurality & +poverty) due to the small volume of cases, which generates some distortions in the graph.

As anticipated by the coefficients of the model, the increase in personal meetings has a more positive effect in the group of municipalities that combine higher levels of rurality with poverty levels below 23.1%. This result is consistent with the previous analysis that used the percentage of the vote as the dependent variable. In fact, personal meetings had a statistically significant effect only in this group. We believe that this effect occurs mainly because in rural communes there is greater proximity between the mayor and his constituents, which facilitates the maintenance of personal links. In most rural communes and those with the largest number of poor people, on the other hand, the positive effect only occurs when the rate of 50 personal meetings per thousand inhabitants is exceeded. In these municipalities longer-term efforts are surely needed to win over voters. The poorest strata are precisely those in which levels of discontent with the parties fester longest, so the task of strengthening loyalties and gaining the trust of voters takes longer to bring results. Finally, in the richest municipalities, personal meetings have a positive and statistically significant effect, but they have rapidly diminishing marginal returns. This is due to the simple fact that in these municipalities, the incumbents have held office for longer periods

and enjoy a more stable and loyal electorate. For this reason, the chances of re-election are greater in this group than in the other groups.

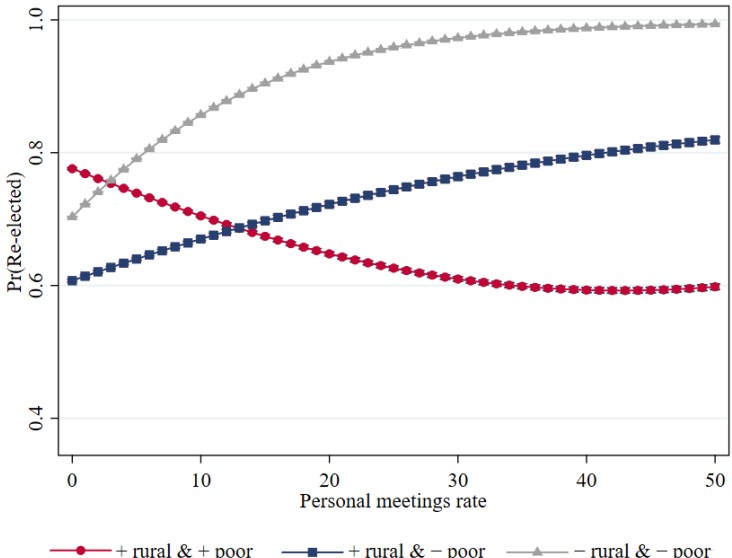

**Figure 3.** Predicted probability of re-election based on the personal meetings rate (by municipality group). Source: Prepared by the authors with data from www.servel.cl and www.infolobby.cl (accessed on 21 March 2021).

Finally, and to test Hypothesis 3, we built four linear regression models whose purpose is to measure the relationship between the granting of personal meetings and levels of electoral turnout (see Table 4). Our hypothesis is that the higher the personal meetings rate, the greater the turnout. This is because the non-programmatic link, whether set up through mediation networks or via direct interaction with the mayor, not only benefits those candidates electorally who adopt these practices, but also mobilises voters who would not have gone to the polls without this clientelist incentive.

**Table 4.** Predictors of the percentage of turnout in the 2016 local elections.

| | Total | (1) +Rural & +Poor | (2) +Rural & −Poor | (3) −Rural & +Poor | (4) −Rural & −Poor |
|---|---|---|---|---|---|
| Personal meetings | 0.821 *** | 0.383 ** | 0.322 | −3.148 | 0.449 |
| | (0.179) | (0.170) | (0.431) | (2.647) | (0.798) |
| Personal meetings | −0.00843 *** | −0.00367 | −0.00981 | 0.696 | 0.0187 |
| | (0.00260) | (0.00223) | (0.0133) | (0.415) | (0.0578) |
| Personal meetings | $1.97 \times 10^{-5}$ *** | $8.66 \times 10^{-6}$ | $7.80 \times 10^{-5}$ | −0.0265 | −0.000324 |
| | $(6.80 \times 10^{-6})$ | $(5.55 \times 10^{-6})$ | $(9.54 \times 10^{-5})$ | (0.0165) | (0.000757) |
| No. of terms in office | 0.349 | 0.114 | 1.923 * | 1.438 | 0.139 |
| | (0.443) | (0.923) | (1.072) | (2.578) | (0.621) |
| Party (1 = N. Majority/0 = Other) | −1.268 | 4.094 * | 3.597 | −1.850 | −1.280 |
| | (1.186) | (2.162) | (2.527) | (3.066) | (1.739) |
| Incumbent expenditure | 0.218 *** | 0.174 ** | 0.208 *** | 0.470 *** | 0.190 *** |
| | (0.0372) | (0.0721) | (0.0639) | (0.113) | (0.0559) |
| Increase in expenditure | 0.0222 | −0.0408 | −0.0429 | 0.0219 | −0.00134 |
| | (0.0264) | (0.0432) | (0.0585) | (0.124) | (0.0395) |
| Difference in expenditure | −0.0842 *** | −0.105 * | −0.101 ** | −0.276 *** | −0.0523 |
| | (0.0296) | (0.0598) | (0.0428) | (0.0844) | (0.0454) |
| Constant | 22.24 *** | 34.25 *** | 29.25 *** | 18.37 *** | 21.83 *** |
| | (2.177) | (5.248) | (4.712) | (5.500) | (3.141) |
| Observations | 264 | 86 | 51 | 23 | 104 |
| R-squared | 0.323 | 0.217 | 0.319 | 0.813 | 0.227 |

Standard errors in parentheses. *** $p < 0.01$, ** $p < 0.05$, * $p < 0.1$. Source: prepared by the authors with data from www.servel.cl and www.infolobby.cl (accessed on 21 March 2021).

Our models show that the rate of personal meetings granted by mayors has a positive and significant impact on turnout in municipal elections. It is probable that in those

municipalities in which the mayor is more inclined to meet with citizens, interest in politics as a public activity will be greater. While a mayor who is accessible and diligent in responding to the requests of neighbours encourages electoral participation, a mayor who does not pay attention to the requirements of his constituents drives voters away from the polls. The overall model is in line with the cubic significance we obtained in some previous models. Replicating the joint significance F-test (0.019), we find that the meetings are also significant for the case of Group 1 communes. Finally, neither the political capital of the mayor, as measured by the number of terms served in office, nor the political coalition to which the mayor belongs are significant predictors of electoral turnout.

To visualise these effects, according to the results of these models, we built a simulation for each group of municipalities. It is worthwhile to note that the group that originally had 24 cases (combining rurality below 32% and poverty above 23.1%) rose to 31 due to the exclusion of the campaign spending variable. As we explained, this variable is not available for every municipality, so omitting it increases the number of valid cases for analysis. Figure 4 shows the results. The effect of the personal meetings is similar for the four groups of municipalities, although with evident variations in the levels of electoral turnout expected. In any case, audiences have a linear effect on turnout, although with quickly diminishing marginal returns.

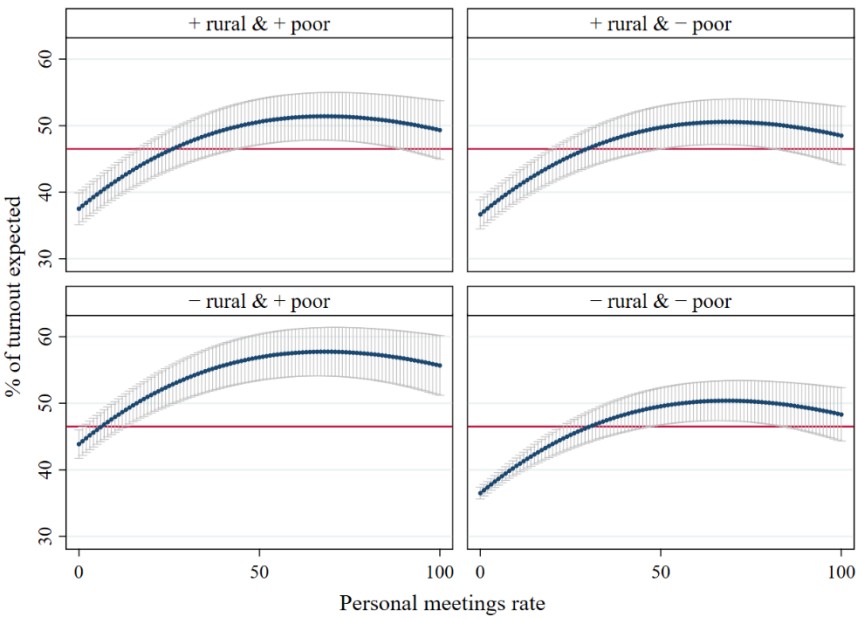

**Figure 4.** Predicted values of electoral turnout according to personal meetings rate (by group of municipalities). Source: prepared by the authors with data from www.servel.cl and www.infolobby.cl (accessed on 21 March 2021).

Finally, a finding that more sharp-eyed readers might notice is the place of poverty in the relationships we find. In principle, clientelist practices should be more likely to occur in areas with high rates of poverty and to decrease as the needs of voters are less pressing. However, our results show that the opposite is the case in less vulnerable municipalities. The explanation for this is that non-programmatic personal meetings are not always granted for the distribution of social benefits. While in the poorest and most rural districts, requests are for social assistance, work or access to education, in the richest ones, personal meetings operate as a mechanism for direct access to decision makers. When we review the details of the requests, for example, we find that several personal meetings in the rich districts were for the presentation of projects to the mayor, and not for social needs.

## 7. Conclusions

We studied the relationship between non-programmatic linkage and electoral performance and success, as well as the effect of such linkage on electoral turnout at the local level. We sought to demonstrate that, apart from the classic determinants developed in the theory, factors exist that influence electoral behaviour and success. These are usually ignored by the literature given their informality and the difficulty of measuring their effects. Yet scholars have pointed out using ethnographic methods that non-programmatic linkages tend to affect turnout and that they improve electoral performance. However, quantitative studies have focused on electoral spending, vote buying and the co-optation of organisations, leaving aside the impact of personalised linkage and the role of political intermediaries.

A study of the personal meetings held by mayors with constituents is a novel way of measuring non-programmatic linkage. Although the main problem in operationalising this variable is the informality of these practices, the new scenario that has emerged in Chile with a public record of all personal meetings granted opens new paths for the investigation of mediation and brokerage. What was an informal practice belonging to the classic grey areas of politics has transitioned into a more institutionalised one in which formal channels are used to attend to the public. The literature has begun to address this type of linkage in formal and/or institutionalised contexts, identifying certain key mechanisms or actors to turn to within government departments.

In general terms, the determinants of electoral success in the 2016 local elections were factors that have long been studied in the history of the literature, notably candidates' performance in the previous elections and the campaign spending of both incumbents and challengers, to name just two of them. However, according to our evidence, certain spending variables, such as the difference in expenditure between the winner and the runner-up, do not have a significant and determining effect. These results are probably explained by the incumbents' strength in their municipalities. Once mayors manage to establish themselves in their town halls, it is quite difficult for challengers to dislodge them.

Non-programmatic linkage, meanwhile, is also relevant in explaining levels of electoral turnout and performance, especially in rural municipalities. Our first hypothesis asserted a relationship between the personal meetings rate and the percentage of the vote achieved. As we were able to verify statistically, this hypothesis is correct. In effect, personal meetings are a predictor of electoral turnout and performance, but we encountered nuances depending on the municipality's sociodemographic profile. Thus, while this statement is true in municipalities with a high rate of rurality and a low rate of poverty, the effect of personal meetings on the percentage of the vote is subject to diminishing marginal returns. This means that granting personal meetings helps mayors obtain a higher percentage of votes. However, increasing the number of personal meetings over the average of 40 personal meetings per 1000 inhabitants does not bring better results.

A challenge of this type of formal mediation study is to learn about the beneficiaries in greater depth. Although asymmetric relationships between patron and client are usually studied, some practices occur within these personal meetings that involve a narrower gap between the actors. In the case of requests for professional work in a municipality, for example, a certain clientelism is developed towards middle-income groups. In the case of municipal permits and commercial patents, the beneficiaries clearly belong to more affluent classes.

Finally, it is most important that subsequent studies measure both the determinants of personal meetings and their effects. Although the manner in which personal meetings are used has been irregular, this may be due to the limited time of their implementation. It is to be expected that as the personal meetings mechanism attracts greater public attention, the bodies in charge of supervising the correct use of the tool will punish mayors who abuse it. For this reason, an analysis of audiences as formal mediation mechanisms may be essential in explaining the success of certain political parties and leaders, as well as the percentages of electoral turnout in the municipalities.

**Author Contributions:** Conceptualization, M.M. and F.B.; Data curation, F.B.; Formal analysis, M.M.; Funding acquisition, M.M.; Methodology, M.M.; Project administration, M.M.; Visualization, F.B.; Writing—original draft, M.M. and F.B.; Writing—review & editing, M.M. and F.B. All authors have read and agreed to the published version of the manuscript.

**Funding:** This research was funded by FONDECYT: 1220004 and Millennium Nucleus Center for the Study of Politics, Public Opinion and Media in Chile: NCS2021_063.

**Institutional Review Board Statement:** Not applicable.

**Informed Consent Statement:** Not applicable.

**Data Availability Statement:** The data were obtained directly from the platform www.infolobby.cl (accessed on 1 July 2022). The authors can be contacted for more information about the filters applied to process the data.

**Conflicts of Interest:** The authors declare no conflict of interest.

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
