# Peer review of "Clientelism, Turnout and Incumbents’ Performance in Chilean Local Government Elections"

_socsci, doi:10.3390/socsci11080361_

Round 1

Reviewer 1 Report

This manuscript aims to measure the relationship between clientelism and electoral performance in Chilean local elections. The authors use a proxy for clientelism that relies on registered meetings between the mayor and members of the public. To the best of my knowledge, the use of this measure is novel, and I think it is very interesting: (if correct) it provides a relatively direct measure of clientelism (or at least of a facet of it), which is something very hard to obtain for quantitative/statistical analyses of clientelism. I think there are multiple areas in which the manuscript could improve significantly, in many cases by clarification. I list my comments in the following paragraphs.

A key issue is the validity of the measure of clientelism that the authors are using: personal meetings with the mayor recorded under Law 20730. The authors should try to provide as much evidence as they can that this measure is indeed capturing clientelism and not something else. My intuition is that the authors are probably right in that higher numbers of “none of the above” meetings signal some kind of clientelistic capture of this institutional mechanism. However, I am not an expert in Chilean politics and institutions, and therefore it is hard for me to determine whether this is really the case here. While I am certainly aware of how other types of institutions and formal mechanisms have been captured in a similar fashion in other countries, the authors should provide evidence that this is a good proxy for clientelistic linkages in Chile. A good first step is the information they cite about the frequency of “none of the above” topics for meetings in election v. non-election years. Other information along these lines could be very helpful. I wish I could suggest specific ways of validating the measure, but since clientelism tends to be based in informal relationships, it is hard to imagine validating ideas without good knowledge of the specific case of Chile.

In terms of the variables used in the analysis, it is not clear why the authors control for difference in incumbent spending between 2012 and 2016 when they’re analyzing the 2016 election. It seems like incumbent spending in 2016 and challenger(s) spending in 2016 are the relevant variables (or some ratio between the two….). Or is this because the authors control for vote share in 2012? Regardless, the spending by opponents should be included one way or another.

I think models should not be weighted by the population of the municipalities since the unit of analysis really is the mayor/office. I would expect the type of clientelistic linkage the authors study to be stronger in smaller constituencies than in the larger ones, thus using this type of weighting would in fact understate the authors’ findings. What I think authors should consider is instead controlling for the population size of the municipalities, perhaps in logarithms (although see subsequent paragraph on rurality) .

Additionally, the authors might consider an interaction model that interacts municipality size with the personal meetings measure to measure whether the effect of personal meetings is stronger in smaller municipalities (or in larger ones) or whether the effect is rather homogenous relative to municipality size. I understand that this is related to what the authors are doing with the split of the sample into groups depending based on the rurality measure, but the two are not really the same. I think what matters here is the size of the municipality, not whether it is urban or rural….Of course, I could imagine that rural municipalities tend to be smaller than urban ones….but are there small urban municipalities as well?

What is the measure of poverty being used?

When the municipalities are broken into groups of high and low rurality and poverty, why do the authors use one standard deviation above the average as a cut-point rather than something like the average or (even better) the median? Either way, this classification should be justified & explained in some way or modified.

Why do the authors use a cubic polynomial for personal meetings? I understand that there may be non-linearities, but the authors should justify the choice of the polynomial degree. Why 3 and not 4 or 2 or 5? In fact, why not a linear model?

When indicating that in most of the models in Table 3 the effect of personal meetings is not statistically significant, is this based on the t-values of the individual meeting coefficients (linear, quadratic, and cubic)? Or is this based on an F-test of collective significance of the three? The proper test should be the F-test, since the three coefficients really come from the same variable. It is possible for the F-test to show significance when the three individual t-test show non-significance (seems unlikely in this case, however).

The discussion in the first paragraph of page 11, directly under Table 3 is somewhat hard to understand and should be revised. Additionally, while the initial increase and then decrease seen in Figure 2 seems statistically significant, the final increase does not seem to be statistically significant and could be simply a consequence of the polynomial of degree 3 the authors chose…a polynomial of degree 3 will have two inflection points by default. Additionally, the authors should not call the initial increase as a “linear relationship” because it is not linear, but cubic. Rather, they may call it “positive” or “increasing” or something along those lines.

The discussion in the paragraph following that is interesting, but speculative (especially about the final rebound), and should perhaps be moved to a discussion or conclusion section rather than mixed with the presentation of the statistical results. Additionally, I am not very confident in the authors’ rationale for the final rebound (increased scrutiny). Is there evidence that, for example, media coverage increases?

It is somewhat surprising that there is a very clear effect of personal meetings on the probability of winning (Table 4) when there seems to be little to no effect on the vote share of the mayors. Normally, the similar literature tends to find effects on vote shares more significant than effects on probabilities of winning (the continuity of vote share helps). Perhaps this is due to the fact that the authors do not control for the vote share in 2012 in the models in Table 4? In any case, this previous vote share should be controlled for in Table 4, since the “electoral strength” in the previous election is clearly an important factor in explaining the changes of reelection that a mayor has and is likely correlated with the intensity with which mayors pursue personal meetings.  

For the results in Table 5, why do the authors include rurality and poverty as control variables rather than split the sample as in the other cases?

Finally, and somewhat repetitively, I think the authors should use the same specifications for all three tables (even if they also include additional specifications in some or all of them that are not present in the other tables). Of course, it makes sense that Table 4 is a probit rather than a linear regression. But why are rurality and povety control variables in Table 5 while they were used for subsetting the sample in 3 and 4? Why is the 2012 vote share included in Table 3 but not in Table 4 when in both cases it is important to control for previous electoral strength? Following a similar logit, Table 5 should control for previous turnout.

Author Response

Dear Sir or Madam, I am pleased to welcome you.

Thank you very much for your comments. In the attached document you will find our answers and the corrections you suggested. 

Reviewer 2 Report

The article's strengths lie primarily in the originality of the focus and the precision of your methodology. I believe you need to emphasize the surprises you encountered in your investigation of your hypotheses.  The hypotheses, I think, are not themselves novel, so where you find departures from them requires more comment from you. The investigation of the personal mayoral meetings in Chile: some history would be helpful. Why/when did the formalization of such meetings become viewed as something to be desired?  On the basis of your conclusions, what recommendations might you make regarding, say, reform of this institutional innovation? Is what you found  of any consequence for political inquiry outside Chile?

Author Response

(The authors gave the same response as above.)

Round 2

Reviewer 1 Report

The authors have address most of my comments in a satisfactory way. There are a few minor items that the authors should consider.

First, on weighting: If the unit of observation were individual persons in Chile, then weighting by commune makes sense if some smaller communes are overrepresented and larger ones are underrepresented in the sample. However, the unit of observation here is, as the authors note, the commune, not individual voters. Thus, weighting by commune population size is not necessary nor correct to estimate the impact of meetings on electoral performance. I think there is an argument to keep this weighting by population size if the authors want to scale their estimates by giving a higher importance in the estimation to locations with more population (but please note that this is different than the weighting being necessary because of some sample representativeness issue).

Second, the should consider including some citations to online news articles covering some cases of increased scrutiny to better illustrate their story. I found online some evidence of this, and seeing that really helps follow the author’s argument much better for those without close knowledge of the Chilean political environment.

Finally, there are a few relevant papers in the literature that connect rather closely with the authors work and should also be discussed in their article: 

Lehoucq. 2003. Annual Review of Political Science.

DeLuca, Jones, Tula. 2002. Comparative Political Studies.

Giraudy. 2007. Latin American Research Review.

Lisoni. 2018. Journal of Politics in Latin America.

Nunez. 2018. Electoral Studies.

Lodola. 2005. Desarrollo Economico.

Author Response

Dear Reviewer,

We reiterate our thanks, in the attached file you will find our replies to your comments. 

Best regards!
